# Polymer Grafted Aluminum Nanoparticles for Percolative Composite Films with Enhanced Compatibility

**DOI:** 10.3390/polym11040638

**Published:** 2019-04-08

**Authors:** Chenggong Yang, Chufarov Marian, Jie Liu, Qi Di, Mingze Xu, Yunhe Zhang, Wei Han, Kun Liu

**Affiliations:** 1State Key Laboratory of Supramolecular Structure and Materials, Jilin University, Changchun 130012, China; cgyang15@mails.jlu.edu.cn (C.Y.); 13159568617@163.com (J.L.); diqi1215@mails.jlu.edu.cn (Q.D.); 2Jilin Supercapacitor Engineering Laboratory, College of Physics, Jilin University, Changchun 130012, China; Chufarovmv@rambler.ru; 3International Center of Future Science, Jilin University, Changchun 130012, China; 4International Joint Research Center for Nanophotonics and Biophotonics, School of Science, Changchun University of Science and Technology, Changchun 130022, China; llmingze@foxmial.com

**Keywords:** aluminum nanoparticles, composite film, dielectric materials, percolative system

## Abstract

Aluminum nanoparticles hold promise for highly energetic materials and sustainable surface plasmonic materials. Most of the commercial Al nanoparticles are prepared via a high-throughput electrical explosion of wires method (up to 200 g h^−1^). However, the use of Al nanoparticles produced by an electrical explosion of wires is limited by their micrometer-sized aggregations and poor stability. Here, we use polystyrene with –COOH end-group to graft onto isolated Al nanoparticles and dramatically enhance their colloidal stability in various organic solvents. We further demonstrate that the polystyrene grafted Al nanoparticles can be doped into polystyrene films with high compatibility, leading to enhanced dielectric properties, such as higher dielectric constant, lower dielectric loss, and stronger breakdown strength. Moreover, the composite film can improve the moisture resistance of embedded Al nanoparticles.

## 1. Introduction

Aluminum nanoparticles (Al NPs) have drawn considerable attention over the years as highly energetic materials and more recently for their applications as surface plasmonic materials [1,2,3]. In comparison with Al micrometer-sized powders, Al NPs have larger surface area, which allows for shorter ignition times, lower ignition temperatures, and higher energy output [4,5,6,7]. The replacement of micrometer-sized aluminum with nanoscale aluminum in the propellant matrix of rockets has shown an increase in propellant burning rates by as much as 100% [8,9]. The National Aeronautics and Space Administration (NASA) is investigating the use of Al NPs in solid rocket fuel [10]. In addition, Al NPs have recently been recognized as a promising candidate for sustainable surface plasmonic materials, due to their unique plasmonic properties in both UV and visible range and their greatly reduced cost compared with noble metal counterparts, i.e., Au and Ag [11,12,13,14,15].

Until now, commercial Al NPs with a size range of 50–200 nm have been mainly prepared by a high-throughput electrical explosion of wires (EEW) method (up to 200 g/h) [16]. The Al NPs prepared by EEW possess a large surface energy, however, they are also susceptible to forming micrometer-scale particle aggregations [17]. The majority of the Al NP aggregations can reach up to tens of micrometers in size, which dramatically reduce their surface area, colloidal stability, and process ability. The aggregation of Al NPs prepared by EEW is almost unavoidable and significantly limits their further applications in diverse research and technology fields.

The surface of Al NPs prepared with EEW method forms a thin aluminum oxide shell (typically 2–4 nm) upon exposure to air. The self-passivation of an aluminum particle results in a dense nanoscale insulating Al_2_O_3_ shells outside of the metallic spheres, which allows the electrons in the metallic core to tunnel through it, and thereby the aluminum/polymer composites exhibit a high dielectric constant as a percolation system; Compared with the traditional approach of dispersing ferroelectric ceramic particles in a polymer matrix to achieve high dielectric constant, the percolative approach requires a much lower volume concentration of the filler. Therefore, this type of dielectric material can have high dielectric constant with balanced mechanical properties and good adhesion strength [18]. Various conductive fillers, such as metal [19,20,21,22], carbon nanotubes [23,24,25,26,27,28], carbon black [29] and conducting ceramic powders [30] etc., have been used to prepare polymer/conductive filler composites or three-phase percolative composite systems. However, these composites still could not be considered as effective candidates for embedded capacitor applications because of the accompanying high dielectric loss and low resistance at a high filler loading level due to the conductive nature of the filler.

During the past few years, much effort has been dedicated to creating dielectric materials possessing high dielectric constant and low dielectric loss. Fabrication of all-organic composites with conductive polymer fillers, such as polythiophenes, polypyrrole, polyaniline, poly (p-phenylene vinylene) metallophthalocyanines, etc. [31,32,33,34,35] is one major route to obtain high dielectric constant material with low loss. Another candidate is nanodielectric, that is, a new class of dielectric material with nanofillers in a polymer matrix for improving interfaces of the filler and matrix and specific electrical properties [36,37,38]. Aluminum nanoparticles are proved to be suitable for nanodielectric, since the nanoscale insulating Al_2_O_3_ shells outside of Al NPs can restrict the electron transfer between Al NPs, thus leading to a very low loss of the Al NPs doped composites [39]. On the other hand, uniform dispersion of nanoparticles in the nanocomposites is required because aggregation of particles inside the polymer matrix leads to deteriorated electrical or dielectric properties (high dielectric loss and bad dielectric strength). To overcome these issues, polymer brushes grafting on the NPs can efficiently stabilize nanoparticles against aggregation, and form well dispersed nanocomposites when the polymer brush is compatible with the polymer matrix [40,41,42,43,44]. Meanwhile, the incorporation of Al NPs into hydrophobic polymer films is a strategy for the protection of Al NPs [45,46]. Under prolonged exposure to air with relative humidity of >85%, moisture can damage the surface passivation oxide layer, leaving the metal Al core highly vulnerable to oxygen and moisture [47].

In this work, we demonstrate that the surface grafting of polymers can dramatically enhance the colloidal stability of Al NPs in various organic solvents, such as toluene, THF, and DMF. The isolated Al NPs grafted with –COOH end-group terminated polystyrene (PS) were uniformly doped into PS films with high compatibility. Compared with aggregated Al NPs (*agg*-Al NPs), the isolated Al NPs grafted with PS-COOH (*iso*-Al NPs@PS) possess thinner oxide layers, which makes the nanocomposite film doped with *iso*-Al NPs@PS closer to a percolative system, thus leading to even higher dielectric constants. Meanwhile, the dramatically enhanced compatibility of nanocomposite film resulted in lower dielectric loss and lower frequency dependence. Specifically, a relatively higher breakdown strength was maintained, which can substantially provide high energy density dielectric materials. The nanocomposite film can also improve the moisture resistance of embedded Al NPs.

## 2. Materials and Methods

### 2.1. Chemical and Materials

All reagents were used as received without further treatment if not otherwise stated. Styrene (99.0%, Xilong Chemical Co., Ltd., Shantou, China), 4,4′-Azobis (4-cyanovaleric acid) (ACVA) and 2,2′-Azobisisobutyronitrile (AIBN) were purchased from Aladdin (Shanghai, China). Toluene was purchased from Beijing Chemical Reagent Co., Ltd., China, and was distilled prior to use. Aluminum wire with 0.2 mm in diameter and of 99% purity was purchased from the Hongjintai Company in Shenzhen, China.

### 2.2. Preparation of Al Nanoparticles

Al NPs in this experiment were produced by electrical explosion of wires (EEW) method [48,49]. Briefly, under argon atmosphere, an initial Al wire with 0.2 mm in diameter and of 99% purity was continuously fed by a feeding roller installed between the high-voltage electrodes. The storage capacitance of the EEW facilities was 96 μF, and the electrodes voltage prior to discharge varied between 4.0 and 4.4 kV. With the high-density current (100 kA), the Al wire was superheated and eventually exploded to form Al vapor that was subsequently condensed to form NPs with diameters in the range of 50−200 nm. The Al NPs were collected in a chamber under Argon atmosphere.

### 2.3. PS Grafting to Al NPs

PS with –COOH end-group was synthesized by free radical polymerization initiated by 4,4′-Azobis (4-cyanovaleric acid) (ACVA) in toluene. ACVA is one kind of azo-initiators which can provide –COOH end-group for polymers. In detail, 2.0 mL styrene monomer with 10.58 mg ACVA were put in 23.0 mL toluene to make the total solution volume 25.0 mL. After being thoroughly degassed, and refilled with N_2_ for 3 times, the system was kept under 70 °C overnight. The grafting of PS-COOH onto Al NPs was conducted in toluene solution, 1.0 mg Al NPs and 1.0 mg PS-COOH were mixed in 2.0 mL toluene solution and ultrasonic for 1 h, then the mixed solution was left standing overnight. The solution was purified by three times of centrifugation in toluene to remove the free PS.

### 2.4. Preparation of the PS Film Doped with Al NPs

To prepare a PS nanocomposite thin film, 1.0 mL toluene solution of PS tethered Al NPs (1.0 mg mL^−1^) were mixed with 1.0 mL toluene solution of PS (2.0 mg mL^−1^) which was prepared by free radical polymerization initiated by AIBN and possessed a *M*_n_ of 21 kg mol^−1^ and PDI of 1.14. The mixture solution was ultra-sonicated for an hour before solvent casting on a quartz substrate (2 × 2 cm^2^). The final PS nanocomposite film doped with isolated Al NPs grafted with PS (*iso*-Al NPs@PS) was baked at 80 °C for 90 min to remove the solvent completely. The nanocomposite film containing aggregated Al NPs (*agg*-Al NPs) was prepared under identical conditions.

### 2.5. Humidity Assay

Examination of the humidity resistance of the PS film was conducted under rigorous conditions. In detail, a quartz substrate with the PS film was placed on the 80 °C water vapor within 2 cm. In this case, high humidity and high temperature can attack the PS film at the same time. We used the UV-Vis spectrometer to monitor the transmittance spectrum of the PS film doped with Al NPs to see the reaction of Al NPs. This process was sustained for 4 h and spectra were taken every 30 minutes. The film had been kept under 80 °C water vapor for over 24 h before the last spectrum was measured.

### 2.6. Measurement of Dielectric Constant

Dielectric constant and dissipation factor of the PS film doped with Al NPs was measured using an Agilent LCR meter (4294A) instrument. This PS nanocomposite film was prepared by solvent casting of Al NPs/PS mixed solution on a quartz wafer. A thin layer of Cu electrode of specific area was evaporated on the film. The measurement was conducted using sinusoidal voltage of peak value 1 V, at frequencies from 10^6^ to 10^2^ Hz.

### 2.7. Measurement of Breakdown Strength

The dielectric breakdown strength of polymer nanocomposite films was obtained by using a high voltage generator and the point contact method. The maximum voltage which the film withstands before the failure took place was recorded as the dielectric breakdown voltage. Each sample was measured five times to obtain the final average values. The breakdown strength was analyzed using a Weibull probability failure analysis method.

### 2.8. Characterization and Instruments

Transmission electron microscopy (TEM) images and high resolution TEM (HRTEM) images were obtained by using a transmission electron microscope (TEM, JEM-2100F, Tokyo, Japan) operated at 200 kV on carbon-coated TEM grids. Scanning electron microscopy (SEM) images was recorded by HITACHI SU8020 (Tokyo, Japan) operated at 3 kV. Dynamic light scattering (DLS) measurement was performed by using a Zeta Sizer instrument (Nano ZS, Malvern Instruments Ltd., Malvern, UK) at 25 °C. X-ray Photoelectron Spectroscopy (XPS) measurement was performed by using a PREVAC XPS/UPS System (PREVAC, Upper Silesia, Poland, Etching of the Al NPs was performed by Ar plasmon). Gel permeation chromatography (GPC) measurement was performed by using an Agilent Technologies 1260 infinity (Agilent Technologies Co. Ltd., Santa Clara, CA, USA) at 35 °C. THF was used as the eluent with an elution rate of 1.0 mL min^−1^, and polystyrene standards were used for calibration. The transmittance spectra were measured by a PerkinElmer Lambda 950 UV-vis-NIR spectrometer with a data interval of 2 nm in the same quartz cell.

## 3. Results and Discussion

### 3.1. Preparation and Characterization of Al NPs Doped PS Composite Films

Figure 1a,c shows the representative TEM and SEM images of Al NPs prepared by EEW method, respectively. The NPs have a broad size distribution ranged from 45 nm to 180 nm as determined by TEM analysis (Appendix A). The Al NPs possessed a core-shell structure with an amorphous oxide passivation outer layer (thickness: ca. 3.5 nm) and a partially crystalline metallic Al inner core as shown in the high-resolution transmission electron microscopy (HRTEM) image of Appendix A. The Al@Al_2_O_3_ core-shell structure was further confirmed by X-ray photoelectron spectroscopy (XPS) combined with plasma etching (Appendix A).

During the EEW process, the as-synthesized Al NPs were aggregated into large conglomerates due to their high surface energy. The conglomerates contained tens to hundreds of Al NPs and showed irregular 3-dimensional (3D) structures (Figure 1a,c). The aggregations were not colloidally stable in various organic solvents with different polarities, such as ethanol, toluene, and acetone. For example, upon sonication in toluene for half an hour, although a suspension was obtained and showed an ash-grey colour due to the strong scattering of large conglomerates, the Al NPs were almost completely precipitated to the bottom of a vial in a few minutes (inset in Figure 1a).

The grafting of polystyrene with –COOH end-group (Appendix A) onto isolated Al NPs was performed in a toluene solution. During the grafting process, Al NPs were uniformly dispersed in the toluene solution, and no sedimentation was observed even after 48 h (inset in Figure 1c). This result suggested that the colloidal stability of the Al NPs was dramatically improved. To remove any physically adsorbed polymers and impurities, the Al NPs were purified by multiple centrifugations. Both the mean hydrodynamic diameter (*D_h_*) and PDI of the Al NPs determined by DLS were remained unchanged during the centrifugation process, further confirming the excellent colloidal stability of the PS grafted Al NPs (Appendix A). This result also indicates the PS was covalently grafted on the surface of Al NPs [50]. The successful grafting was further maintained by FTIR spectra in Appendix A, and the grafting density was calculated by the TGA (Appendix A) to be 0.12 chains/nm^2^.

Figure 1b,d shows the representative TEM and SEM images of the isolated Al NPs after being grafted with PS. The Al NPs formed 2-dimentional random close-packing on the TEM grid. Clear inter-particle spaces (ca. 4.5 nm) can be observed between the NPs, corresponding to the PS grafted on their surface. DLS results in Appendix A show that the *D_h_* of aggregated EEW Al NPs and PS grafted isolated Al NPs were ca. 1 µm, and 200 nm, respectively, similar to the sizes of Al NPs determined by TEM analysis. Close inspection of the TEM images of Figure 1a,b reveal that compared with the original aggregated Al NPs, the shape of the isolated NPs became more angular. In addition, the HRTEM image of the isolated Al NPs in Appendix A shows that the amorphous oxide passivation layer became thinner and indiscernible. These results suggest that the Al oxide layer was “etched” by the –COOH end-group of PS. The reaction between PS-COOH and Al NPs surface was further confirmed by the XPS result in Appendix A.

After the grafting process, isolated Al NPs (*iso*-Al NPs@PS) were doped into PS films to fabricate dielectric nanocomposite films. As a control, PS films doped with aggregated Al NPs (*agg*-Al NPs) were also prepared. Figure 2a,b shows the optical microscope images of PS films doped with the two kinds of Al NPs respectively. In Figure 2a, it can be seen that the *agg*-Al NPs formed large aggregation “islands” with sizes of tens of micrometers in the PS film. In contrast, iso-Al NPs@PS were dispersed very well in the PS film, which shows a much more uniform contract as shown in Figure 2b. It can be concluded that the compatibilities of these two composite films were significantly different. The difference of the PS films doped with isolated and aggregated Al NPs was attributed to two factors: one is the size effect, and the other is the PS chains grafted on the surface of Al NPs, which facilitate the mixing of Al NPs with the PS matrix [51]. (Related TEM images of can be seen in Appendix A)

The fracture surfaces of the PS films dopes with Al NPs for series filler loadings were observed by SEM (scanning electronic microscope), as shown in Figure 3. It can be seen that large voids existed in the composite film with *agg*-Al NPs, which should deteriorate the dielectric properties. In contrast, *iso*-Al NPs@PS were doped in the composite films much more uniformly than that of *agg*-Al NPs, regardless of their filler loading.

The hydrophobic PS film can offer moisture resistance to the Al NPs. The moisture resistance test of the polymer films was conducted to compare the chemical stability of the Al NPs under an 80 °C water vapor treatment. In the case of the PS film doped with *agg*-Al NPs as shown in Appendix A, the transmittance of the film was quickly decreased due to the formation of Al(OH)_3_ under 80 °C water vapor treatment for up to 1 h, and then increased to nearly 100% by 2 h as the produced Al(OH)_3_ had almost completely fall off from the film. In contrast, the film doped with *iso*-Al NPs@PS showed an excellent moisture resistance property (Appendix A). The initial transmittance was about 10% lower than that of the film doped with *agg*-Al NPs, indicating the *iso*-Al NPs@PS were uniformly dispersed in the film. The transmittance changed by only 1% even after 80 °C water vapor treatment for 24 h.

### 3.2. Dielectric Properties of Al NPs Doped PS Composite Films

Figure 4a,b shows the dielectric constants of nanocomposite films doped with *agg*-Al NPs and *iso*-Al NPs@PS, respectively, in comparison with that of a pure PS film. With the frequency changing from 10^2^ to 10^6^ Hz, the dielectric constant of pure PS film remained about 2.80, which was consistent with the reported PS dielectric constant of 2.40–2.60 [52]. Compared with the pure PS film, the PS films doped with *iso*-Al NPs@PS or *agg*-Al NPs showed higher dielectric constants due to the passivation oxide shell. The insulating Al oxide shell of *agg*-Al NPs was ca. 3.5 nm, which was thick enough to provide insulating properties. In contrast, *iso*-Al NPs@PS possessed thinner oxide shells (ca. 1.5 nm), which made the nanocomposite film doped film closer to a percolative system, thus leading to even higher dielectric constants. For the films doped with *agg*-Al NPs (Figure 4a), the dielectric constant was frequency dependent with the filler loading being increased (Appendix A), which was mainly due to the poor compatibility of the *agg*-Al NPs in the PS film, leading to the considerable interfacial polarization. As interfacial polarization is usually strong at low frequencies [53], the frequency dependent phenomenon was more obvious in the low frequency range. Moreover, *agg*-Al NPs with their filler loading of 50 wt % were no longer compatible with the PS film (Appendix A). In contrast, the filler loading of *iso*-Al NPs@PS can be up to 50% in the doped PS film. The dielectric constant of the doped film proportionally increased with the filler loading of Al NPs, as shown in Figure 3b, resulting in a significant increase of the dielectric constant to 12.0 for the Al NP filler loading of 50%. This dielectric constant of 12.0 is significantly higher than that of 7.79 for the Al_2_O_3_-doped PS films reported previously [54]. More importantly, the dielectric constants of the doped films were almost frequency independent, which should be attributed to the weak interfacial polarization between the PS films and the doped Al NPs.

The dissipation factor of nanocomposite films doped with *agg*-Al NPs and *iso*-Al NPs@PS were present in Figure 4c,d, respectively. It can be seen that the dissipation factor of pure PS film appears under 0.015 from 10^2^ to 10^6^ Hz. After doped with *agg*-Al NPs, the dissipation factor slightly raised to no more than 0.03. Although adjacent nanoparticles were aggregated together, the insulating oxide layer restricted the electron transfer between aluminum particles, thus leading to a very low loss of the nanocomposites. The irregular variation of dissipation factor should be attributed to the poor compatibility. The dissipation factor of films doped with *iso*-Al NPs@PS was lower than that of *agg*-Al NPs. Dissipation factors were below 0.02 for most of the frequency range, and, changed more regularly owing to the good compatibility of isolated Al NPs (Appendix A).

Breakdown strength (*E*_B_), which decides the energy density of the dielectric material, is another important parameter of dielectric materials [55]. The breakdown strength was measured in order to evaluate energy densities of the nanocomposite films doped with different types of Al NPs. The *E*_B_ were analyzed by a two-parameter Weibull statistical distribution method and the result is shown in Figure 5. The nanocomposite films doped with *iso*-Al NPs@PS and *agg*-Al NPs show significantly different *E*_B_ values. The characteristic *E*_B_ values of the pure PS film are about 233 kV mm^−1^. The *E*_B_ values of the films doped with *agg*-Al NPs were significantly reduced compared to the pure PS film. When the filler loading was 10 wt %, the breakdown strength was dramatically dropped from 183.77 kV mm^−1^ at the filler loading of 10% to ca. 30 kV mm^−1^ at the filler loading of 20 wt % and 30 wt % (Appendix A). This result was mainly caused by the poor compatibility of the films doped with *agg*-Al NPs. When the voltage across the composite film becomes too high, the passivation Al_2_O_3_ shell can no longer withstand the high voltage, and the composite film begin to conduct current. Under this condition, electrons can transfer between neighboring particles, the aggregation particles then formed current access and defected easily, which dramatically reduced the breakdown strength of the composite film. In contrast, the films doped with *iso*-Al NPs@PS showed good compatibility, due to well-dispersity of *iso*-Al NPs@PS in the PS film. The *E*_B_ was preserved in the range of 211~175 kV mm^−1^ and decreased slightly with the filler loading of Al NPs. (Appendix A)

The maximum storage energy density (U) of the nanocomposite films doped with *iso*-Al NPs@PS and *agg*-Al NPs were calculated according to equation (1) [56]:(1)$$U=\frac{1}{2}\varepsilon_{0}\varepsilon E^{2}$$
where *ε* is the dielectric constant of the composites and *E* is the breakdown strength. Figure 6 displays the calculated maximum storage energy densities of the nanocomposite films at power frequency (10^3^ Hz). For the nanocomposite films doped with *agg*-Al NPs, the maximum storage energy densities with all Al filler loadings were lower than that of pure PS film, which was caused by the low breakdown strength of the composite films. For the nanocomposite films doped with *iso*-Al NPs@PS, their maximum energy densities were increased with the Al filler loadings and became higher than that of PS film, owing to the enhanced dielectric constant and relatively high breakdown strength of the composite films.

Figure 7 shows the dependence of the alternating current (AC) conductivity as a function of the frequency of the PS film doped with *agg*-Al NPs and *iso*-Al NPs@PS grafted with PS chains. Compared with pure PS film, the composite film doped with *agg*-Al NPs had a lower electrical conductivity. This result shows that aggregated Al NPs with 3.5 nm oxide shell were more insulating than PS film. However, the films doped with *iso*-Al NPs@PS exhibited slightly higher electrical conductivity than pure PS films, which was caused by the thinner oxide shell and high conductivity of metal Al. Because the oxide shell of *iso*-Al NPs@PS was thinner than that of *agg*-Al NPs, the film was closer to a percolative system. However, the isolated Al NPs-PS composite films are still highly insulating, since the electric conductivity of the composite exhibits strong dependence on the frequency [57]. The conductivity did not vary with the filler loading of Al NPs, which suggests the similarity of the polarization property between Al NPs and PS film.

## 4. Conclusions

In summary, we have reported the successful fabrication of Al NPs-polystyrene composite film. Owing to the surface grafted–COOH PS, isolated Al NPs can be better doped into PS films than naked aggregated Al NPs, thus improving the compatibility of the composite film, almost no voids are present in the film, which further enhanced the dielectric constant and maintained the breakdown strength of the PS film. Moreover, compared with other dielectric materials, the dielectric constant of isolated Al NPs doped PS film was almost frequency independent, which may provide new application prospects in this area. At the same time, the PS film can provide moisture resistance for the embedded Al NPs. We believe that this method can be a widely applicable approach for fabricating dielectric composite materials possessing high energy density combined with low dielectric losses. The realization of the fabrication of low-cost Al-NP doped dielectric films with enhanced dielectric constant and low dielectric loss significantly broadened the scope of large scale applications for super capacitors, microwave devices, photoelectric devices and power capacitors of the electronic fields as well as the motor cable.

## Figures and Tables

**Figure 1 polymers-11-00638-f001:**
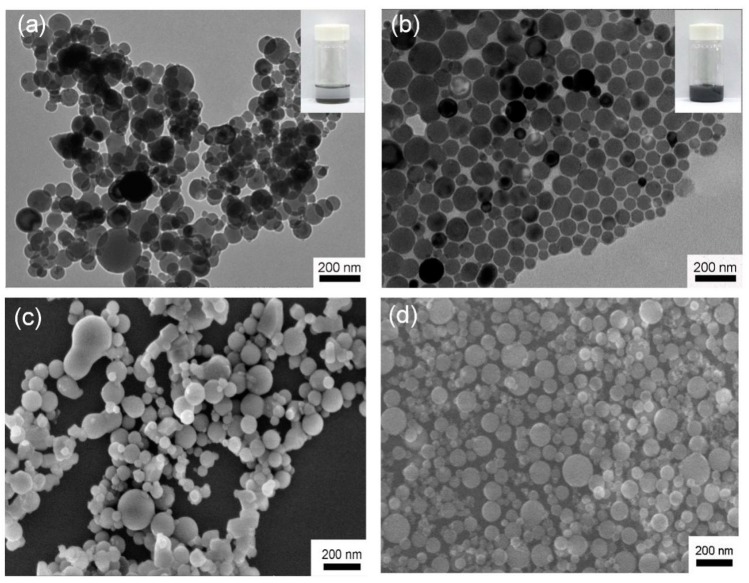
TEM (**a**) and SEM (**b**) images of aggregated Al NPs prepared by EEW method and TEM (**c**) and SEM (**d**) isolated Al NPs after grafting with polystyrene with –COOH end-group, respectively.

**Figure 2 polymers-11-00638-f002:**
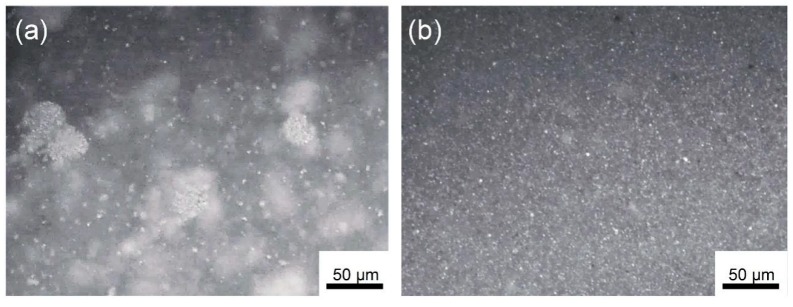
Optical microscope images of (**a**) PS film doped with *agg*-Al NPs and (**b**) PS film doped with *iso*-NPs@PS.

**Figure 3 polymers-11-00638-f003:**
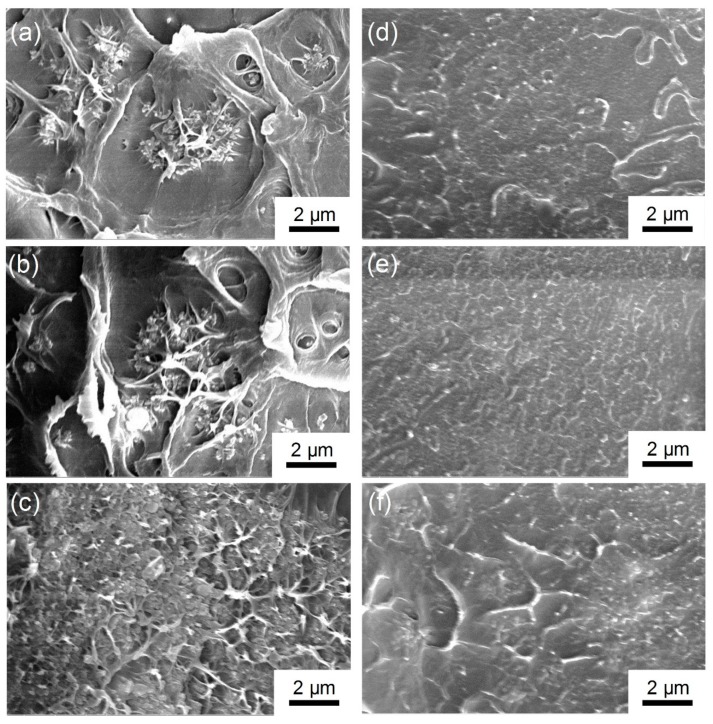
SEM images of the fracture surface of the nanocomposites doped with *agg*-Al NPs (**a**) 10 wt %, (**b**) 20 wt %, and (**c**) 30 wt % and with *iso*-Al NPs@PS (**d**) 10 wt %, (**e**) 20 wt %, and (**f**) 30 wt %.

**Figure 4 polymers-11-00638-f004:**
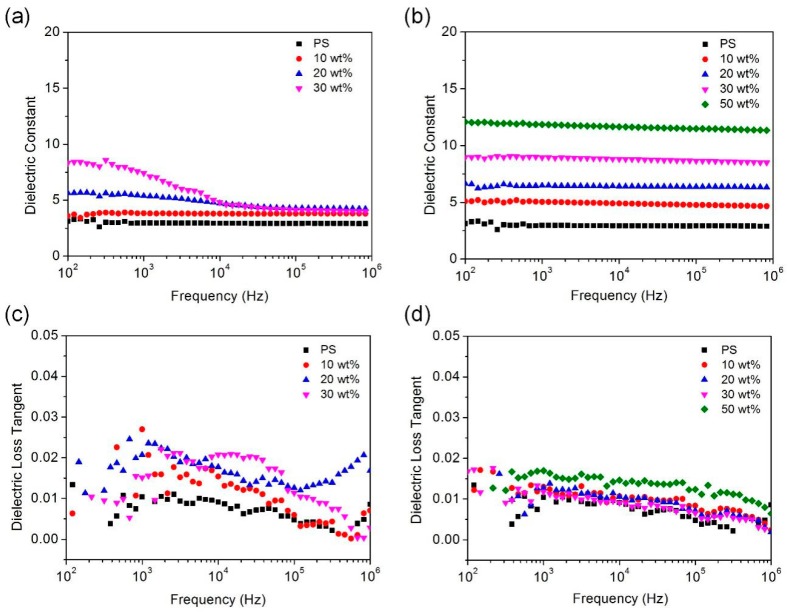
Dielectric constant of pure PS films and Al/PS composite films (**a**) PS film doped with *agg*-Al NPs and (**b**) PS film doped with *iso*-Al NPs@PS. Dissipation factor of pure PS films and Al/PS composite films (**c**) PS film doped with *agg*-Al NPs and (**d**) PS film doped with *iso*-Al NPs@PS.

**Figure 5 polymers-11-00638-f005:**
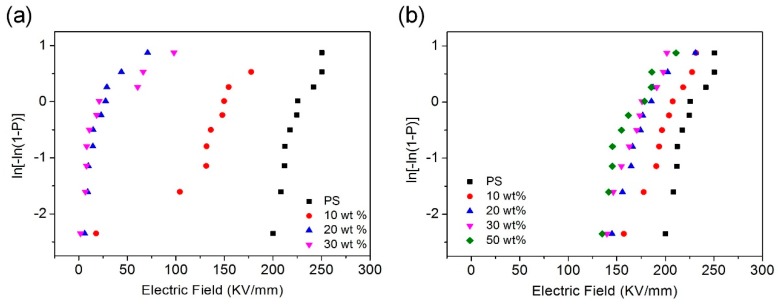
Breakdown strength of Al/PS composite films (**a**) PS film doped with *agg*-Al NPs and (**b**) PS film doped with *iso*-Al NPs@PS.

**Figure 6 polymers-11-00638-f006:**
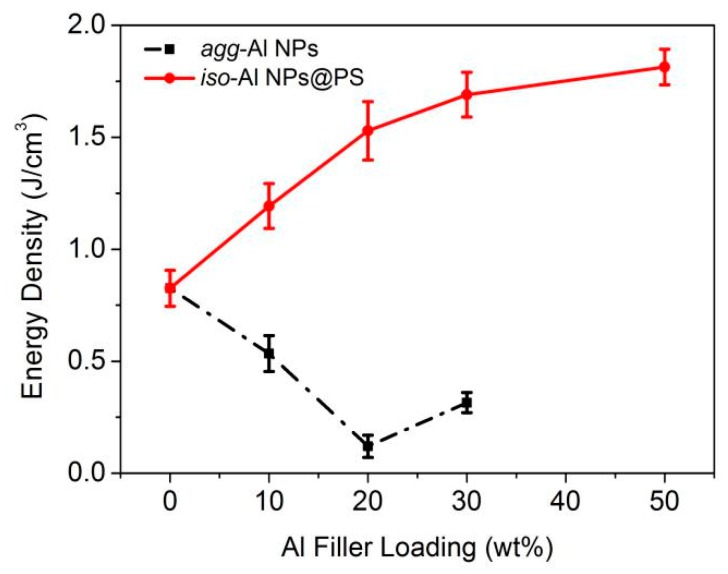
The calculated maximum energy storage density of the nanocomposite films doped with *agg*-Al NPs and *iso*-Al NPs@PS.

**Figure 7 polymers-11-00638-f007:**
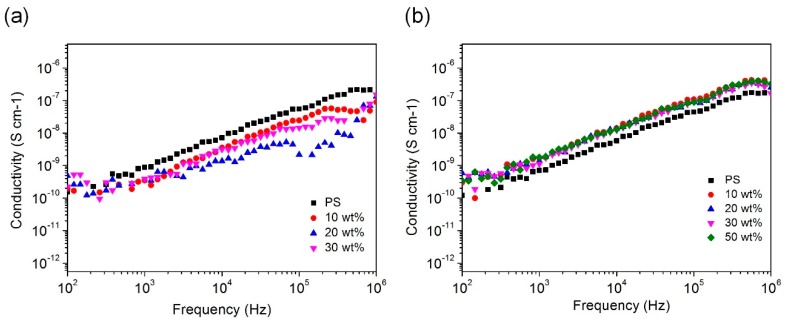
Frequency-dependent electrical conductivity of Al/PS composite films (**a**) PS film doped with *agg*-Al NPs and (**b**) PS film doped with *iso*-Al NPs@PS.

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
