# Peer review of "Polymer Grafted Aluminum Nanoparticles for Percolative Composite Films with Enhanced Compatibility"

_polymers, 2019, doi:10.3390/polym11040638_

Round 1
Reviewer 1 Report
In this study, the authors developed polystyrene grafted Al nanoparticles. The authors also showed the polystyrene grafted Al nanoparticles can be doped into polystyrene films with high compatibility, leading to enhanced dielectric properties. This work is interesting. However, revision is still recommended.
1. Major concerns: what are the potential applications of those films? In addition, Although the nanocomposite doped film showed better property than that of the pure PS films, the author should discuss and compare the property of their nanocomposite doped films to that of other films with nanocomposites published in literature in the discussion part. For instance, paper titled with "Preparation and application of polystyrene-grafted alumina core-shell nanoparticles for dielectric surface passivation in solution-processed polymer thin film transistors" Furthermore, Discussion part should not be combined with conclusion part but can combine with result part. The author should have separate conclusion part.
2. What is the grafting density or thickness of the PS brushes on the nanoparticles?
3. For figure 2, can author provide TEM images to show the well dispersion of nanoparticles within the films
4. Please provide FITR of nanoparticles before and after grating and label the major PS peaks.
5. For figure 6, please show the standard deviations.
Author Response
The point-by-point response to the reviewer'su comments are in the file uploaded.

Reviewer 2 Report
In the manuscript titled “Polymer Grafted Aluminum Nanoparticles for Percolative Composite Films with Enhanced Compatibility” the authors report a method to prepare polystyrene films homogeneously doped with aluminum nanoparticles. In general, the manuscript is well written and structured. However, some revisions are needed before its publication.
1) The successful PS grafting on the surface of Al nanoparticles must be confirmed by using FTIR and/or TGA. 2
2) Page 5: “These results suggest that the Al oxide layer was “etched” by the –COOH end-group of PS.”
Authors should provide more explanation on how the COOH groups can etch the oxide layer. Also, author should provide additional evidence to prove that oxide layer was etched after PS grafting. For this, authors could run XPS experiments as they did for uncoated Al nanoparticles (Fifure S1d)
3) Authors should give the details of PS-COOH synthesis and characterization, and grafting the Al nanoparticles with that polymer in the section 2.3.
4) Page 8:” The EB was preserved in the range of 211~175 kV 266 mm-1 and did not change with the filler loading of Al NPs. (Figure S6b and Table S1)”
As it can be seen from Figure S6b, the EB decreases with the increaning nanoparticle loading. Please correct this statement.
Author Response

(The authors gave the same response as above.)

Round 2
Reviewer 1 Report
The authors addressed my comments. I don’t have any more comments.
Reviewer 2 Report
Authors have addressed my comments in the revised manuscript and it can be accepted in present form.